# Population Structure of *Pyrola chlorantha* (Family Ericaceae) at the Southern Range Margin (Samara Region, Russia)

Valentina Ilyina [1], Stepan Senator [2,*], Anna Mitroshenkova [1], Olga Kozlovskaya [3] and Ivan Kazantsev [1]

1   Faculty of Natural Geography, Samara State University of Social Sciences and Education, 443090 Samara, Russia
2   Tsitsin Main Botanical Garden, Russian Academy of Sciences, 127276 Moscow, Russia
3   Institute of Oil and Gas Technologies, Samara Polytech Flagship University, 443100 Samara, Russia
*   Correspondence: stsenator@yandex.ru

**Abstract:** The population structure of endangered species is one of the main criteria for assessing their state in their habitats. Representatives of the Ericaceae family are sensitive to environmental changes, including anthropogenic pressure; thus, they are considered the indicator species in assessing phytocenose stability. The population structure and density of the threatened species green-flowered wintergreen, *Pyrola chlorantha* Sw., have been described at the southern range margin (south-east of the European part of Russia, Samara Region). The observations were performed here in 2006–2021, and the main parameters of the age and spatial structure of *P. chlorantha* populations were revealed for the first time. Green-flowered wintergreen populations were studied at monitoring study sites and at temporarily established study plots. A bush part (ramet) was set as a counting unit. In total, 27 sub-populations were surveyed, with 1520 individuals registered. The age structure of populations was characterized using common demographic indicators: the recovery index and the population age index. The age structure of the population was associated with the efficiency of both vegetative and seed reproduction. Generally, the share of pre-generative individuals was 32.3%, generative, 66.9%, and senile, 1.8%. The studied populations were stable due to low anthropogenic impact at the growth sites.

**Keywords:** Buzuluksky Bor National Park; individuals; age structure; spatial structure; threatened species; Zhiguli State Nature Reserve

## 1. Introduction

Currently, large-scale anthropogenic pressure on ecosystems causes significant changes in the structure and organization of natural complexes, including vegetation cover (communities, plant populations) [1–7]. At the same time, rare plant species (mostly stenobionts) disappear from plant communities first, so they are included in the lists of protected plants [8–10].

The protection of rare species is impossible without knowledge of their bioecological characteristics. In this regard, population-age methods are relevant and serve as the basis for assessing the current state of vulnerable plant species populations [11–15]. Monitoring of threatened plant populations is one of the main approaches in conservation biology and plant ecology [16–20]. These populations are studied as a part of the monitoring of rare species included in the Red Lists [21–24].

Many authors pay special attention to the structural and functional features of plant species and their populations near the range margin [25–31]. A targeted study of plant populations helps to identify factors limiting the natural succession of plant populations and to propose appropriate conservation measures.

For a long time, both the biology and ecology of *Pyrola* species attracted much attention [32–38]. As a rule, Palaearctic range of green-flowered wintergreen, *Pyrola chlorantha*

Sw., coincides with that of *Pinus sylvestris* L. Green-flowered wintergreen is less moisture-demanding in comparison with other representatives of the genus *Pyrola*, so it serves as an indicator of dry soils [39].

Despite the fairly wide distribution of *P. chlorantha* in Russia, the populations of this species have been studied quite rarely. There are some data on the population structure for the Mordovia State Nature Reserve [40] and the Volzhsko-Kamsky State Nature Reserve [41], but they are insufficient to assess the current population state in natural habitats. In Eastern Europe, the populations of green-flowered wintergreen have been studied in the Desnyansko-Starogutsky National Park (north-eastern Ukraine), where rather stable seed reproduction by plants was observed [42].

The Samara Region (Russia) is the southern range margin of *P. chlorantha* [43,44]. Here, green-flowered wintergreen is rare due to the small number of suitable habitats and rather high anthropogenic load [44]. No studies of the spatial-age structure of *P. chlorantha* populations in Samara Region have been previously conducted.

The study aims to describe the spatial and age structure of *P. chlorantha* populations at the southern range margin (Samara Region, Russian Federation), assess the dynamics of age structure and total population size, and search for the main regularities of the spatial distribution.

## 2. Materials and Methods

### 2.1. Species Description

Green-flowered wintergreen *Pyrola chlorantha* is found in Scandinavia, Atlantic, Middle and Eastern Europe, the Mediterranean, the Caucasus, Western and Eastern Siberia, the Far East, Asia Minor, and North America [43–45].

*P. chlorantha* is a herbaceous, evergreen, rhizomatous perennial, mesophyte, sciophyte, and mycotroph. It grows in pine (*Pinus sylvestris* L.) and pine-broadleaved forests. In the Samara Region, total species abundance is not high, population density is low; and the specimens grow in groups, usually not more than 5–15 individuals per 100 m$^2$ [44].

The peculiarities of the life form of green-flowered wintergreen are well described [7,46]. Most species of the Ericaceae family are usually represented by a bushy part; in *P. chlorantha*, the latter develops for 5–8 years. The life span of the main module is approximately 10 years. In green-flowered wintergreen, all axillary buds of one of the elementary shoots are often initiated into growth simultaneously [46].

### 2.2. Study Area

The Zhiguli State Nature Reserve and the Buzuluksky Bor National Park are federally protected areas within the borders of the Samara Region. The main forest types, where *Pyrola chlorantha* has been registered are lichen-green-moss pine forests and tall grass pine forests (Buzuluksky Bor National Park), pine forests, and oak (*Querqus robur* L.) forests (Zhiguli State Nature Reserve). The Zhiguli State Nature Reserve is located in the Pre-Volga area in the forest-steppe zone (53.405796° N, 49.779342° E). The climate is moderately continental. Average annual precipitation is about 580 mm. The area is represented by a thick formation of limestones and dolomites of Carboniferous and Permian ages. The bedrock, apart from the overlying clay loam layer, is often covered with rubbly clastic weathering products up to 5–20 cm thick.

The Buzuluksky Bor National Park is located in the Samara Trans-Volga Region in the steppe zone (53.011025° N, 51.982755° E); the climate here is also moderately continental. Average annual precipitation is 530 mm, but its uneven nature predetermines extreme moisture conditions, manifested either in extreme drainage of the territory and lowering of the groundwater level or in significant humidification. The soils are sandy and loamy-sandy loam.

In general, habitat conditions are characterized by sufficient aridity, mosaic grasses and tiers, and sparse tree cover. Anthropogenic pressure in the form of fires, recreation,

and grazing is often present in the study area, thus affecting the structure and dynamics of *P. chlorantha* populations.

### 2.3. Experimental Design

*P. chlorantha* populations were studied during the growing seasons of 2006–2021 in accordance with standard methods [5,47,48]. The populations were searched using the route method and then studied. The main techniques of this method were direct observation, assessing the state of a sub-population, measuring and describing plants, and plotting the diagrams and maps. Populations were studied at monitoring and temporarily established sites (from 10 to 150 m$^2$). The population characteristics included the population size, the total area covered, the absolute number of individuals and their population density, the actual contour of the phytocenoses, and the degree of anthropogenic transformation of the phytocenoses.

The age structure of populations was characterized using standard demographic indicators: the recovery index [49] and the population age index [6,50]. The population state was estimated using the "delta-omega" criterion of Zhivotovsky [51]. The replacement index (Irep.), recovery index (Irec.), index of aging (IAg), age index (Δ), and index efficiency (ω) are primary indicators of age structure dynamics in plant populations [47–51].

A bush part (ramet) was set as a counting unit. The number of leaves in the rosette was counted, the length and width of the leaf plates were measured, the height of the flower stalk, and the number of flowers were determined for generative plants.

The age state of individuals of *P. chlorantha* was assessed with regard to that of the closely related species *P. rotundifolia* L. [52].

### 2.4. Statistical Analysis

The life states of the registered individuals were specified, the age structure was assessed, and the main demographic parameters of the populations were calculated based on the measured morphological parameters of individuals. The analysis was performed using PAST 4.04 software (Øyvind Hammer, University of Oslo, Oslo, Norway) [53] and Microsoft Excel 2010 (Microsoft, Redmond, WA, USA).

## 3. Results

Fifteen populations of *P. chlorantha* in the Zhiguli State Nature Reserve and twelve populations in the Buzuluksky Bor National Park were surveyed. The total number of recorded individual plants was 1520:980 in Zhiguli State Nature Reserve and 540 in Buzuluksky Bor National Park, respectively. The average number of individuals (aboveground shoots) in the populations of the Zhiguli State Nature Reserve was ~65 specimens (from 20 to 84); in the Buzuluksky Bor National Park, 45 plants (from 33 to 79). The peculiarities of the population structure of green-flowered wintergreen, important for environmental planning, have been described for the first time.

The age structure of the studied populations of *P. chlorantha* differed in two surveyed geographical locations. In the Zhiguli State Nature Reserve, the age structure was right modal, with a predominance of old generative plants (34.0%). In the Buzuluksky Bor National Park, the spectrum was characterized as bimodal, with the predominance of virginal individuals (33.7%). In both cases, the populations included all age groups (Figure 1). In the Zhiguli State Nature Reserve, a smaller share of young and mature generative plants was noted than that of old generative plants, which might indicate a gradual suppression of the population. The Buzuluksky Bor National Park population exhibited a basic age structure that formed under optimal habitat conditions.

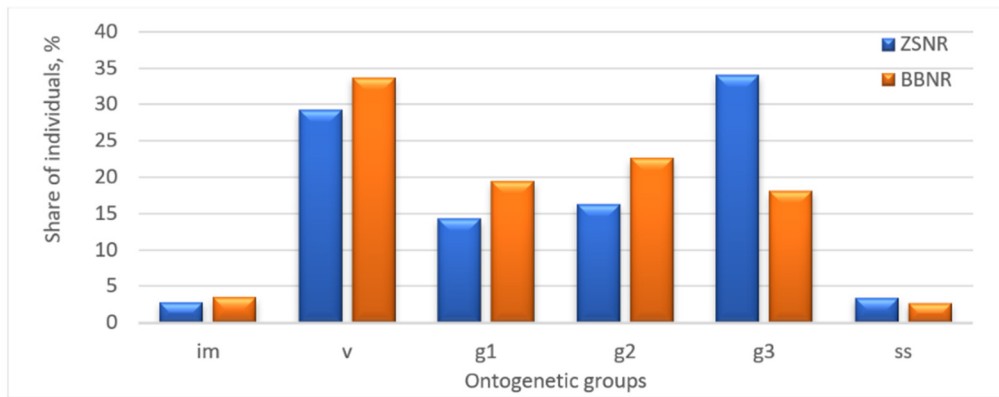

**Figure 1.** Age structure of *P. chlorantha* populations in the Zhiguli State Nature Reserve (ZSNR) and in the Buzuluksky Bor National Park (BBNR): im—immature; v—virginal; g1—young generative; g2—mature generative; g3—old generative; ss—sub-senile individuals (compiled by the authors).

The dynamics of the age structure of *P. chlorantha* populations were worth analyzing separately in the Zhiguli State Nature Reserve and in the Buzuluksky Bor National Park (Figures 2 and 3). In order to estimate the composition and dynamics of populations, we chose the data for the years with the highest number of recorded individuals.

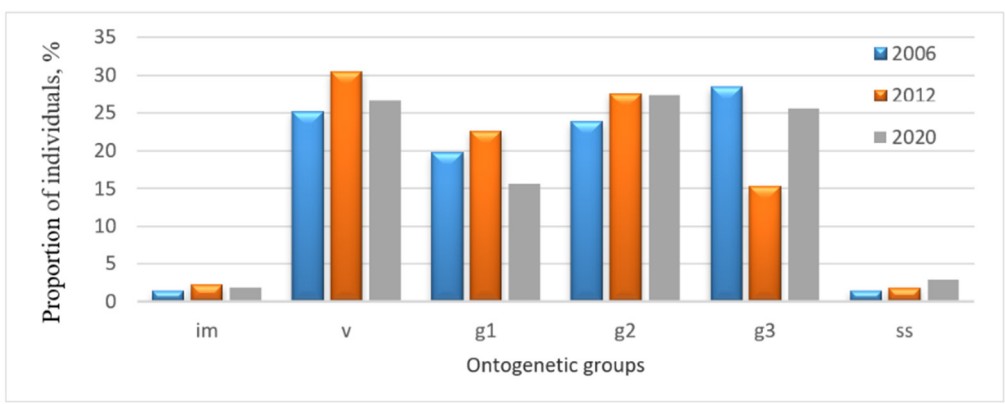

**Figure 2.** Age structure dynamics of the *P. chlorantha* population in the Zhiguli State Nature Reserve: im—immature; v—virginal; g1—young generative; g2—mature generative; g3—old generative; ss—sub-senile individuals.

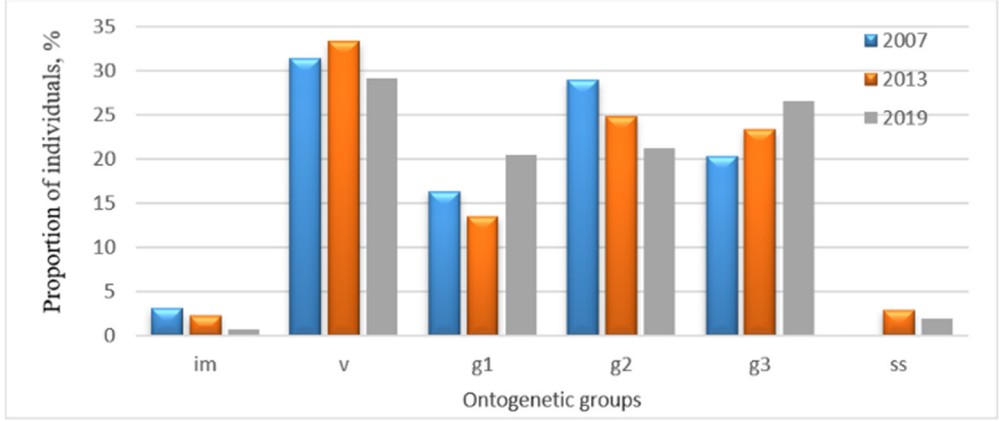

**Figure 3.** Age structure dynamics of the *P. chlorantha* population in the Buzuluksky Bor National Park: im—immature; v—virginal; g1—young generative; g2—mature generative; g3—old generative; ss—sub-senile individuals.

Pronounced dynamics of the age structure were observed in the Zhiguli State Nature Reserve. From 2006 to 2012, the group of old generative individuals experienced the greatest change, with their share nearly halving, from 28.5% to 15.3% (Figure 2).

In the Buzuluksky Bor National Park, the dynamics of the age structure of the population was close to normal. However, the proportion of mature generative individuals decreased (from 28.9% to 21.2%), but the share of old generative plants has increased (from 20.3% to 26.6%).

The age structure of populations and referring indexes allowed for some assumptions (Table 1). Generally, pre-generative individuals represented about a third of the population (32.3%), generative individuals made up two-thirds (66.9%), and senile individuals made up a minor portion(1.8%). The population indices were: (1) the index of replacement (IRep) of 0.46; (2) the population recovery index (IRec) of 0.47; (3) the aging index (IAg) of 0.02; (4) the ageing index ($\Delta$) of 0.40; and (5) the efficiency index ($\omega$) of 0.71. According to the delta-omega criterion [48,51], the populations of both Buzuluksky Bor National Park and the Zhiguli State Nature Reserve belonged to the mature type, but transitional populations were noted in this area in some years. The studied parameters differed significantly (Student's *t*-test, tSt = 2.08), but did not differ statistically according to the nonparametric test ($p < 0.05$).

**Table 1.** Age structure of populations of *P. chlorantha* at the southern range margin (ZSNR—the Zhiguli State Nature Reserve, BBNP—the Busuluksky Bor National Park): p–v—pre-generative plants; g1–g3—generative plants; ss–s—sub-senile and senile individuals.

| Protected Area | Year | Share of Age Classes in the Population, % | | | Indices | | | | |
|---|---|---|---|---|---|---|---|---|---|
| | | p–v | g₁–g₃ | ss–s | $I_{Rep}$ | $I_{Rec}$ | $I_{Ag}$ | $\Delta$ | $\omega$ |
| ZSNR | 2006 | 26.5 | 72.1 | 1.4 | 0.36 | 0.37 | 0.01 | 0.42 | 0.73 |
| | 2012 | 32.7 | 65.5 | 1.8 | 0.49 | 0.50 | 0.02 | 0.36 | 0.71 |
| | 2020 | 28.6 | 68.5 | 2.9 | 0.40 | 0.42 | 0.03 | 0.42 | 0.72 |
| BBNP | 2007 | 34.5 | 65.5 | 0 | 0.53 | 0.53 | 0 | 0.38 | 0.71 |
| | 2013 | 35.5 | 61.6 | 2.9 | 0.56 | 0.58 | 0.03 | 0.40 | 0.69 |
| | 2019 | 29.8 | 68.3 | 1.9 | 0.42 | 0.44 | 0.02 | 0.41 | 0.71 |

The spatial structure of *P. chlorantha* populations was similar at all studied sites due to the peculiarities of its vegetative reproduction in the Samara Region. In each population, there was a cluster formed by 10–50 individuals. The distance between clusters was from 1.5 to 30 m in both the Zhiguli State Nature Reserve (2006, 2012, 2020) and the Buzuluksky Bor National Park (2007, 2013, 2019) (Table 2). Cluster formation was predominantly due to the vegetative reproduction and mosaic nature of the grass layer. The average population density was calculated for the plots of 1, 10, and 100 m². Single individuals or the small groups of two or three specimens (mostly individuals of seed origin) were found very rarely between the clusters. However, in most cases, other *P. chlorantha* plants were not registered between the clusters.

**Table 2.** Spatial distribution of individual *P. chlorantha* at the southern range margin (ZSNR—the Zhiguli State Nature Reserve, BBNP—the Busuluksky Bor National Park).

| Protected Area | Year | Average Population Density | | | Number of Plant Individuals per Cluster | Distance between Clusters, m * | | |
|---|---|---|---|---|---|---|---|---|
| | | per 1 m² | per 10 m² | per 100 m² | | min | M ± m | max |
| ZSNR | 2006 | 12.3 | 22.6 | 67.3 | 11.7 | 0.5 | 1.5 ± 0.6 | 33.5 |
| | 2012 | 6.8 | 12.4 | 37.3 | 5.7 | 0.8 | 1.9 ± 1.1 | 39.6 |
| | 2020 | 11.2 | 16.4 | 42.5 | 10.4 | 0.7 | 1.8 ± 0.4 | 39.2 |
| BBNP | 2007 | 14.5 | 34.6 | 65.9 | 24.7 | 1.6 | 3.4 ± 1.1 | 22.6 |
| | 2013 | 12.7 | 33.7 | 56.7 | 25.1 | 1.6 | 3.9 ± 0.7 | 22.6 |
| | 2019 | 13.4 | 30.8 | 54.1 | 17.4 | 1.8 | 3.9 ± 0.8 | 23.4 |

* M—mean, m—error of mean, min—minimum value, max—maximum value.

Vegetative reproduction served as the main way of replenishment in the studied populations of *P. chlorantha*, as evidenced by the high share of virginal individuals: 29.2% in the Zhiguli State Nature Reserve and 33.7% in the Buzuluksky Bor National Park. A small number of immature plants of vegetative origin (2.8% and 3.5%, respectively) was recorded as well. In general, the individuals that appeared from the seeds accounted for about 3% of the total population in the Zhiguli State Nature Reserve and about 4.5% in the Buzuluksky Bor National Park.

In some years, forest fires led to the deterioration of *P. chlorantha* populations. Large forest fires were recorded in Zhiguli State Nature Reserve in 2010 and in Buzuluksky Bor National Park in 2010 and 2021.

## 4. Discussion

The predominance of vegetative reproduction over seed reproduction in *P. chlorantha* is observed in both the Zhiguli State Reserve and the Buzuluksky Bor National Park. During vegetative propagation, new young individuals are characterized by an elongated plagiotropic shoot; the individuals emerged from the seeds as orthotropic rosette shoots. Low shares of seed reproduction in the Zhiguli State Nature Reserve may be explained by the peculiarities of the limestone substrate and the weak development of the humus layer [54]. Low rates of *Pyrola* seed reproduction have been reported earlier [55]. Low potential seed productivity affects the morphogenesis of individuals, abundance, renewal processes, and the age structure of many rare plant species populations [56–59].

The green-flowered wintergreen population in the Buzuluksky Bor National Park is characterized by better renewal due to replenishment by immature and virgin individuals. The species that reproduce vegetatively usually have a high potential for recovery [60–62]. However, the number of individuals in the studied populations of green-flowered wintergreen is low, indicating a low level of recovery associated with the elimination of plants in the early stages. Most often, the limiting factors for population growth are anthropogenic pressure [63] and the ecological and coenotic conditions of habitats [64]. In the Zhiguli State Nature Reserve, green-flowered wintergreen populations gradually accumulate generative individuals that become the dominant group after some time. The accumulation of individuals in the generative group is typical for plant species with a long life span [63,65].

Numerous publications repeatedly emphasize the rather narrow ecological and coenotic niches of representatives of the Ericaceae family and the pronounced effect of climate changes on their bioecological characteristics [66,67]. The presence of mycorrhizal fungi is indispensable for seed germination and further plant growth [27,68]. Differences in the environmental conditions of *P. chlorantha* habitats in the Zhiguli State Nature Reserve and the Buzuluksky Bor National Park are reflected in the population structure.

The ageing of the *Pyrola* population in the Buzuluksky Bor National Park, as manifested by a gradual increase in the share of old generative individuals, is most likely due to recreational impact on vegetation cover, which is influenced by changes in ecological

and coenotic conditions reported in different regions [69]. However, the total, absolute abundance of green-flowered wintergreen has not undergone strong changes. A decrease in the number of individuals in clusters and an increase in the distance between them are associated with partial disturbance of the ground cover in some areas due to timber activities and cattle grazing.

The age structure dynamics of green-flowered wintergreen in the Zhiguli State Nature Reserve are close to normal for the species. The decrease in the share of old generative plants is associated with the disturbance of the soil and vegetation cover during the catastrophic forest fires of 2010, which have significantly affected the oak and pine forests of the reserve [70]. After the fires of 2010, the number of green-flowered wintergreen individuals in the studied area have also decreased significantly, but the species began to recover its numbers subsequently.

Forest lowland fires reduce the vitality and abundance of *P. chlorantha* and cause changes in the spatial and age structure of populations, resulting in a decrease in the number of individuals in clusters and an increase in the distance between the preserved plant clusters. This is due both to the elimination of existing individuals and the severity of affecting the surface soil layer. Fire exposure often leads to the elimination of the mycelium of the fungus mycorrhiza associated with *P. chlorantha*. According to some data, recovery of mycorrhizal fungi in soil takes a long time [71–73], which also affects population growth, age structure, and population state in general. Comprehensive monitoring is a prerequisite for assessing the fire effects on the populations of rare plants [74].

## 5. Conclusions

The age structure of *P. chlorantha* populations indicates that they are more stable in the Buzuluksky Bor National Park compared to those from the Zhiguli State Nature Reserve. The plants are characterized mainly by vegetative reproduction, predetermining the specimen clusters in plant communities as well as the predominance of pre-generative individuals in populations. The basic spectrum of the left modular type was revealed in the *P. chlorantha* populations in the Buzuluksky Bor National Park. This type is characteristic of many long-rhizome plant species. The differences in the age structure of populations in the Zhiguli State Nature Reserve are apparently related to the features of the stony substrate, which complicate both vegetative and seed reproduction. At the same time, the populations are characterized by lower rates of self-renewal, so the emergence and successful settlement of young individuals are difficult.

**Author Contributions:** Conceptualization, V.I. and S.S.; methodology, V.I.; data acquisition and analysis, V.I., A.M., O.K. and I.K.; writing—original draft preparation V.I., A.M. and S.S.; writing—review and editing, V.I. and S.S. All authors have read and agreed to the published version of the manuscript.

**Funding:** This research was funded by Institutional research project No. 122042700002-6 of the Tsitsin Main Botanical Garden, Russian Academy of Sciences. We thank the Ministry of Science and Higher Education of the Russian Federation for their support of the Center of Collective Use "Herbarium MBG RAS", grant 075-15-2021-678.

**Institutional Review Board Statement:** Not applicable.

**Informed Consent Statement:** Not applicable.

**Data Availability Statement:** Not applicable.

**Conflicts of Interest:** The authors declare no conflict of interest.

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
