# Peer review of "Population Structure of Pyrola chlorantha (Family Ericaceae) at the Southern Range Margin (Samara Region, Russia)"

_2037-0164, doi:10.3390/ijpb13040051_

Round 1

Reviewer 1 Report

p. 18, 77 , 116.Replace " partial shrub partial" with " partial bush partial"

 p. 85, 87, 93. Replace square brackets with round ones

 р. 104. Incorrect link. Need  “Coenopopulations of Plants: Essays of Population Biology”, Moscow: Nauka, 1988

 p. 111. Invalid link. This index was first proposed by A.A. Uranov (1975)

 p. 112-115.  Link to literary sources

 p. 141. This spectrum is bimodal (see Coenopopulations of Plants, 1988; Zaugolnova, 1994)

 p. 23-24.  Numbers  32.1%, 65.8%, 2.1% do not match the averages in Table 1

numbers 0.48; 0.49 do not match the average in the table  one.

 p. 180-182.  This is mistake. 5 coenopopulations belong to the mature type (see Zhivotovsky, 2001;  Osmanova, Zhivotovsky, 2020), one coenopopulation (Buzuluksky pine forest) is transitional (see Zhivotovsky, 2001;  Osmanova, Zhivotovsky, 2020).

 p. 268-269. The left-modal spectrum is not characteristic of all long-rhizome plants. In some long-rhizomatous species, the spectrum is centered (see Smirnova, 1987; Zaugolnova, 1994) and bimodal (for example Potentilla bifurca, Basargin, 2011; Cheryomushkina, Basargim, 2011).

Author Response

Many thanks to the reviewer for the thorough work with our article. We have made corrections. Please see the attachment

Reviewer 2 Report

Line 11-12: “Structure of plant populations is not clear here. What do you mean by structure here?

Line 32: Citation needed

Line 32-34: These two sentences can be summarized in one and be clearer. Please rewrite.

Lines 37-39: Rewrite by: “Monitoring of threatened plant populations is one of the main methods to evaluate their conservation status”

Line 39-40: This sentence is not clear. The study of populations is indeed what monitoring consists of. I would say “The monitoring of rare species populations is ongoing for those included in the Red Data Books in the Russian Federation [14-17]”

Line 45: The topic of the Introduction has changed drastically at this point. I would smooth the transition from conservation biology to Pyrola chorology/ecology

Line 48: Scientific name in italics

Line 50: “[…] populations of the species have been rarely studied.

Line 60: This must be rewritten: “The research tasks were to determine the population dynamics considering ontogeny, total size and the main regularities of the spatial distribution of individuals.

Paragraph 2.1.: I feel this is quite repetitive with the content of Lines 49-59. I suggest to compile this information in a single section.

Lines 67-70: citation needed. Also, is it necessary to write all the regions?

Line 76: citation needed

Line 83: “The main forest types where Pyrola was registered” – do you mean P. chlorantha or the whole genus?

Lines 93-94: “[…] is also located in a moderately continental climate with an average annual precipitation of 530 mm.”

Lines 92-101: The information provided in this section can be written together instead of presenting isolate sentences apparently independent among them.

Line 100-101: “This influences the structure and dynamics of P. chlorantha populations

Section 2.3. should be entitled “Experimental design”

Line 105: Briefly explain what the route method consist of

Section 2.4. I am afraid that you will need to develop more deep statistics analysis to be publishable on an indexed journal

Line 135: This sentence is not adequate in Results section

MAJOR ISSUE: Figures 1, 2, 3 lacks statistic support. Error bars were made by Excel default system and the aesthetic is not appropriate for a high-level scientific work. Also, significant differences are not assessed by any statistical method anywhere. I consider that this must be deeply improved for further consideration.  I am not revising Results section since this major issue is fixed, because the information can substantially change after statistical analysis.

Line 212: How can you distinguish seed reproduction versus vegetative reproduction using your data?

Line 216: Pyrola in italics

MAJOR ISSUE: I am not deeply revising Discussion since the interpretation of data may vary after statistical analysis.

Author Response

(The authors gave the same response as above.)

Round 2

Reviewer 2 Report

-